# Uterus Transplantation: Revisiting the Question of Deceased Donors versus Living Donors for Organ Procurement

**DOI:** 10.3390/jcm11154516

**Published:** 2022-08-03

**Authors:** Emily H. Frisch, Tommaso Falcone, Rebecca L. Flyckt, Andreas G. Tzakis, Eric Kodish, Elliott G. Richards

**Affiliations:** 1Obstetrics and Gynecology and Women’s Health Institute, Cleveland Clinic, Cleveland, OH 44195, USA; frische@ccf.org (E.H.F.); falcont@ccf.org (T.F.); 2Department of Obstetrics and Gynecology, Division of Reproductive Endocrinology and Infertility, University Hospitals Cleveland Medical Center, Cleveland, OH 44106, USA; rebecca.flyckt2@uhhospitals.org; 3Digestive Disease and Surgery Institute, Cleveland Clinic, Cleveland, OH 44195, USA; tzakisa@ccf.org; 4Pediatric Institute/Cleveland Clinic Children’s and Cleveland Clinic Lerner College of Medicine, Cleveland, OH 44195, USA; kodishe@ccf.org

**Keywords:** uterus transplant, deceased donor, living donor

## Abstract

Uterus transplantation is a surgical treatment for women with congenital or acquired uterine factor infertility. While uterus transplantation is a life-enhancing transplant that is commonly categorized as a vascular composite allograft (e.g., face or hand), it is similar to many solid organ transplants (e.g., kidney) in that both living donors (LDs) and deceased donors (DDs) can be utilized for organ procurement. While many endpoints appear to be similar for LD and DD transplants (including graft survival, time to menses, livebirth rates), there are key medical, technical, ethical, and logistical differences between these modalities. Primary considerations in favor of a LD model include thorough screening of donors, enhanced logistics, and greater donor availability. The primary consideration in favor of a DD model is the lack of physical or psychological harm to a living donor. Other important factors, that may not clearly favor one approach over the other, are important to include in discussions of LD vs. DD models. We favor a stepwise approach to uterus transplantation, one in which programs first begin with DD procurement before attempting LD procurement to maximize successful organ recovery and to minimize potential harms to a living donor.

## 1. Introduction

The field of uterus transplantation (UTx) is expanding both in the scientific literature and in public awareness, as programs worldwide continue to report livebirths resulting from UTx in women with absolute uterine factor infertility (AUFI). It has been a decade since the first successful uterus transplant, and the question of whether to use deceased donors (DDs) or living uterus donors (LDs) is still an area of investigation and debate. This paper examines the relative merits and liabilities of each approach, exploring logistical, surgical, medical, and ethical considerations of both LDs and DDs.

In a DD model of uterus procurement, a uterus is procured from a patient pronounced dead by neurologic criteria along with life-saving organs [1]. In a LD model of UTx, the uterus is procured via hysterectomy, ideally in premenopausal women with proven fertility. The LD procurement is not a simple hysterectomy, as it is similar to a radical hysterectomy in the potential morbidity to the donor.

The LD model continues to be the most common approach in this evolving field [2]. LDs can be categorized by their relationship to the recipient, either as known (usually related to the recipient, such as a mother, but can include unrelated individuals such as in-laws and family friends) or altruistic donors. 

The first UTx in humans in modern times was performed in 2000 using a LD, and the second was performed in 2011 using a DD [3]. Both of these attempts were performed outside of a clinical trial. The first clinical trial of UTx was performed in Sweden; the trial exclusively utilized LDs and had the world’s first live birth in 2014. The world’s first live birth from DD occurred in Brazil in 2017, followed shortly by the Cleveland Clinic [4]. To date, more than 100 procedures have been performed [2]. In 2020, Category III CPT codes for UTx were established in the United States for both LD and DD procurement.

There are multiple active, announced, or completed clinical trials involving DD, LD, or both. DD programs include the Cleveland Clinic (ClinicalTrials.gov identifier (accessed on 25 July 2022: NCT02573415); Ghent University Hospital Women’s Clinic Ghent, Belgium (ClinicalTrials.gov identifier: NCT03252795); and Imperial College London, UK (ClinicalTrials.gov identifier: NCT04244409). (A DD trial was initiated at the University of Nebraska but was ultimately canceled.) LD programs include Sahlgrenska University Hospital Gothenburg, Sweden (ClinicalTrials.gov identifier: NCT03590405); Mansoura Urology and Nephrology Center at Mansoura, Al-Dakahliya, Egypt (ClinicalTrials.gov identifier: NCT03284073); Barretos Cancer Hospital Barretos, São Paulo, Brazil (ClinicalTrials.gov identifier: NCT04249791); Hospital Clinic Barcelona, Spain (ClinicalTrials.gov identifier: NCT04314869); and Hospital Foch Suresnes, France (ClinicalTrials.gov identifier: NCT03689842). Centers that utilize both DD and LDs include Baylor University Medical Center Dallas, Texas, United States; University of Pennsylvania, USA (currently only recruiting for LD trial: ClinicalTrials.gov identifier: NCT03307356); Brigham and Women’s Hospital, USA (not actively enrolling through a clinical trial); Texas Children’s Hospital Houston Texas, USA (ClinicalTrials.gov identifier: NCT05263076); Sahlgrenska University Hospital Gothenburg, Sweden (ClinicalTrials.gov identifiers: NCT03138226, NCT03581019, NCT05089513); and the Institute for Clinical and Experimental Medicine Prague, Czechia (ClinicalTrials.gov identifier: NCT03277430). This information was retrieved in July 2022 from clinicaltrials.gov.

Currently, there are two programs that offer uterus transplant outside of a clinical trial. Baylor Scott and White in Dallas offers LD transplants for an out-of-pocket fee. The University of Alabama performs DD transplants outside of a clinical trial [5,6].

## 2. Discussion

### 2.1. Considerations in Favor of the LD Approach

#### 2.1.1. More Complete Medical History of the Donor

One of the most important advantages of a LD is the ability to obtain a more complete medical history of the donor. The medical history is limited in a DD, where the transplant team must rely on written records or recollections of family members which may be incomplete. In both LDs and DDs, work-up includes ultrasound, pelvic magnetic resonance [7], and microbiology testing to evaluation for infections and structural abnormalities. However, details such as cervical dysplasia history, obstetrical history, or presence of absence of gynecologic complaints, may be missing from the medical record and can only be obtained by interviewing a LD.

#### 2.1.2. LD Availability

DDs in UTx are pronounced dead by neurologic criteria, not donation after cardiac death; DD must retain cardiac activity before procurement. Additional criteria for prospective DDs vary by institution, often requiring a history of full-term birth, absence of any chronic disease, the absence of any conditions that would affect survival of the graft, a BMI less than 30 kg/m^2^, and the structural suitability of the uterus [8]. Furthermore, programs and/or their prospective recipients may opt out of using “donors with risk criteria”, which was established by the US Public Health Services (PHS) in 2020. These criteria greatly limit the volume of DD. Programs also decide the radius of travel that they would allow for a DD uterus.

#### 2.1.3. Clear, Authentic Consent

UTx is classified as a vascularized composite allograft (VCA). VCAs are a subset of life-enhancing organs rather than life-saving organs, and include transplants of the face and hands. As a VCA, procurement requires explicit next of kin authorization. When women sign up to be organ donors, they may not initially think their uterus is included [9]. Preferences regarding tissue donation are often complex; some patients may be less willing to donate their non-vital after death, especially reproductive organs [9]. Organs may only be removed from prospective DDs when there is prior explicit consent or surrogate authorization by family members [10]. A uterus donation from the deceased may raise ethical concerns in some societies and religions. This request may also negatively affect obtaining consent for procuring other life-saving organs, but this has not been reported for DD UTx [11].

The UK NHS Transplant Activity Report demonstrated that 88% are willing to donate all tissues and organs after death whereas 12% are selective and hold strong preferences against specific organs such as heart and cornea. Reassuringly, a more recent survey of a diverse sampling of the US population showed that 74.4% women were willing to donate their uterus [12]. The ethical justification for allowing any of these procedures to be done include comprehensive informed consent.

In the case of LDs, it is critically important to engage in a thorough informed consent process. There are significant risks with the surgical procurement which has higher complexity and morbidity than a typical hysterectomy, making a thorough consent process ethically mandatory. The consent process also includes a detailed explanation that the uterus transplant does not entail parental rights and that relationships with the offspring of a transplanted uterus are not expected and are decided by the recipient [7].

#### 2.1.4. Easier Logistics When Planning the Procurement

LD has considerable logistical advantages. Potential donors can be identified and screened ahead of time, and the procurement procedures can be planned during normal business hours with advance coordination of all care teams. This is in contrast with DD procurements, which often involve unpredictable timing.

The scheduled LD surgery typically happens in an operating room adjacent to the recipient’s operating room, allowing for faster mobilization and less ischemic risk, with the average cold-ischemia time of 1–2 h [13,14]. The location of DD procurement is not guaranteed to be at the same location, so the recipient’s surgery can cause delays in tissue transport and increase in cold ischemia time [13].

#### 2.1.5. Better Histocompatibility When Using Directed Donation 

Directed LD UTx reduces the potential immunologic mismatch that can occur in the DD transplantation, and may also elevate the potential for a genetically related donor–recipient relationship, thereby reducing the risk of other histocompatibility-related issues [15].

### 2.2. Considerations in Favor of the DD Approach

#### 2.2.1. Avoidance of Risk of Physical Harm to the Donor

Unlike with LDs, there is no risk for morbidity and mortality for a DD [11]. The potential harms to a LD are not insignificant: donating a uterus has a rate of 19% for serious postsurgical complications requiring surgical or radiological intervention [2]. Complications include bleeding requiring surgical exploration, ureteral injury, dehiscence of the vaginal cuff, and pain [16,17]. Ureteral injuries occur in about 10% of donors. It is important to note, however, that the complications for LDs have decreased as a center gains more experience [18]. This is similar to the learning curve observed in living donor liver transplants, where significant improvements in operation time, blood loss, and postoperative recovery came with greater volume and experience of a center [19].

In addition, there is an increased risk for ovarian failure and risk of sexual dysfunction after hysterectomy. Even with ovarian conservation, increased cardiovascular and metabolic morbidity in patients has been demonstrated in patients undergoing hysterectomy [20].

There have been cases of oophorectomy during LD procurement. Bilateral oophorectomy causes surgically induced menopause. Whether planned or unplanned, oophorectomy significantly increases all-cause mortality, cognitive function, cardiovascular disease, mood disorders, sexual dysfunction, and osteoporotic fractures [21,22].

#### 2.2.2. Faster Procurement

As is the case with liver and kidney transplants, DD procurement is easier, faster, and technically more straightforward. In the DD model, a large transverse incision (such as the Tzakis incision) allows for removal of the uterus along with the life-saving organs. The uterus is typically retrieved in under 3 h [13]. Unlike with LD procurement, there is no need to preserve retroperitoneal structures, allowing for more generous dissection of vascular pedicles [14].

Uterus procurement typically happens after procurement of vital organs, though there is variability reported in the literature [23,24]. When properly performed, uterus procurement in a DD should not impact procurement of vital organs, though this is a unique concern for DD procurements, which involve vaginotomy in order to remove the specimen.

In contrast, while there is great variability in the length of time of LD procurement reported in the literature, times are significantly longer than in DD, ranging from 6–8 h [25] to 10–13 h [23,26].

#### 2.2.3. Avoidance of Risk of Psychological Harm to the Donor

There are certain expectations that involve donating a uterus, and directed LDs may experience long-term psychological effects if the transplant is unsuccessful [24]. After enduring surgical risk themselves, donors have social and emotional investment in the recipient’s outcomes. Uterus donation is considered successful after a live birth from the transplant, and with it comes different expectations from a donor [27]. LDs should be thoroughly evaluated pre-operatively to identify psychological risk factors or pre-existing psychiatric disorders [11].

#### 2.2.4. Optimal Age of the Uterus

Most LDs, both directed and non-directed, are peri- or post-menopausal at time of donation. In preparation of transplant, post-menopausal donors are often prescribed a course of cyclic estrogen to enhance vessel size prior to donation. This in itself increases risk to the donor, particularly if not already on hormone therapy. The dosage of estrogen in these LD protocols exceeds the amounts that are given in postmenopausal hormone replacement therapy. In contrast, DDs are younger and premenopausal, typically at an age that would be more ethically problematic for a LD.

### 2.3. Considerations for UTx in Which Neither LD nor DD Has a Clear Advantage

#### 2.3.1. Outcomes Data

Outcomes data is the single most important consideration when comparing LDs and DDs. Recent reports from the Czech Republic and the United States seek to address the question as to which type of graft yields better outcomes with respect to the primary objective of uterus transplantation, which is live birth [18,28].While many more hundreds of LD and DD transplants will be needed in order to determine noninferiority of cumulative live birth rates with statistical confidence, the report from the United States provides important preliminary data from multiple institutions that allow us, for the first time, to directly compare outcomes between LD and DD transplants.

While live birth and long-term safety are considered to be the most important indicators of success, “success” in uterus transplantation is broken into seven progressive stages/milestones: (1) technical success, (2) menstruation, (3) embryo implantation, (4) pregnancy, (5) delivery, (6) graft removal, and (7) long-term follow-up [29]. In the aggregate United States data, the following was reported with regards to these milestones:Technical success did not vary between LD and DD attempts, with graft loss being 26% for LD attempts and 24% for DD attempts.Menstruation did not vary between LD and DD attempts, with mean onset of menses being 31 days in LD recipients (95% CI, 24.4–37.1) and 29 days in DD recipients (95% CI, 18.5–40.4; *p* = 0.82).Livebirth rate was reported as having no statistically significant difference between LD and DD recipients, with the aggregate being 83%, or 36% live birth per embryo transfer.The embryo implantation rates, clinical pregnancy rates, miscarriage rates, gestational age at delivery, graft-to-hysterectomy times, and hysterectomy outcomes were provided only in aggregate with no graft-specific data for LD and DD recipients.

In addition, cold ischemia time was increased for DD grafts (332.4 min; 95% CI, 267.3–397.5) compared with LD grafts (219.7 min; 95% CI, 188.2–251.1; *p*  <  0.001), whereas LD grafts had longer mean warm ischemia time (63.3 min; 95% CI, 55.2–71.3) compared with DD grafts (47.5 min; 95% CI, 29.2–65.8; *p*  = 0 .06). The authors speculated that while the longer cold ischemia time may have implications for long-term graft survival, long-term function is less important in the setting of a graft that will be removed, which may be why no difference in clinical outcomes was observed [18].

Besides their preliminary nature, these data must be interpreted with some caution. Success rates and protocols at IVF centers and uterus transplant programs vary. In the United States, the majority of LD transplants were performed at a single institution while the majority of DD transplants were performed at a different institution. In addition, there have been more LD UTx procedures than DD UTx procedures [11].

More research is needed on success rates as programs continue both DD and LD UTx, including comparisons of the complications for recipient surgery and recipient pregnancies [28]. The most common severe complication of UTx is arterial or venous thrombosis. The overall vascular complication rate is approximately 20% and most commonly results in graft hysterectomy [17]. Maternal complications in UTx have been reported as high as 37.5% of transplants, but more data are needed [8,17]. There is a higher rate of preterm birth and respiratory distress syndrome from newborns of transplanted uteri, although a comparison between LD and DD has not been addressed [2]. Long-term data for recipients from LD and DD UTx, as well as for their families and offspring, are greatly needed.

#### 2.3.2. Other Ethical Considerations

Utilizing DDs alleviates many of the ethical challenges inherent in LD UTx, since DDs cannot be harmed physically or psychologically. However, philosophers point out that there is a difference between harming and wronging someone, and it is plausible that a deceased donor could be “wronged” by using her uterus in a transplant if that is something that she would have objected to when alive [30,31,32]. As discussed above, individuals often have complex preferences regarding non-vital organs [9] but that survey data show that most women are willing to donate their uterus after death [12].

With non-directed LDs, the most frequent motivation for donation is the ability to provide another woman the opportunity to carry a pregnancy and have a child [33]. Using a casuistic approach, one might compare living donation of the kidney and liver to living uterus donation. The ethical justification for allowing any of these procedures to be done include comprehensive informed consent and the ethical permissibility of supererogatory acts (actions “above the call of duty” that are morally good but not morally required). While living liver donation is clearly a higher risk procedure than living kidney donation, it could be argued that the commensurate benefit to the donor is greater for liver than kidney because of the availability of dialysis for the latter. By contrast, UTx is a completely elective procedure and the indication of uterine factor infertility is not a life-threatening condition. This would suggest that the risks of living donation of the uterus are not as clearly justified from a donor perspective in contrast to liver and kidney. Despite this difference, a truly informed and voluntary donor should not be prevented from acting in this altruistic way. In addition, like liver donation, there may not be alternate treatments available (such as gestational surrogacy) for the patient with absolute uterine factor infertility. Autonomy to make decisions is dependent on informed consent, which in turn requires understanding of risks, benefits, and alternatives of a procedure. The consent process for directed and non-directed donors is important to screen for coercion and to respect autonomy. The level of comprehension of this highly complex procedure that is deemed sufficient to be a living uterus donor is unclear.

The majority of the LD uterus transplants performed in Europe used the recipient’s mother as the donor [15]. With related/directed donors, the relationship between the recipient and the donor is easier to navigate when the donor spontaneously volunteers to provide their uterus rather than being approached by non-family or medical personnel. This is a two-edged sword, however: while there is potential for an emotional benefit to all family members involved having a sibling or parent offer their uterus, there is also potential for emotional fall-out when a graft is not successful, as was shown in Sweden (matts). When a transplant surgery is not successful, a known LD may feel that they have “failed” the recipient, particularly if the donor is the mother of the recipient and already harbors irrational feelings of guilt over the recipient’s congenital uterine factor infertility. Alternatively, when a donor surgery has complications, the recipient may likewise feel guilty and responsible for this poor outcome for the donor.

A system of uterus procurement that relies exclusively on known LDs is unjust, as patients with absolute uterine factor infertility without a willing known potential LD would be unable to enroll in a trial or receive the treatment. In this regard, a system of anonymous LDs or deceased donors would be preferable. Similarly, distributive justice is also a concern in the DD model, as the number of patients desiring UTx far exceeds the number of potential DDs, as discussed above.

#### 2.3.3. Cost Considerations

The costs for a LD vs. DD in UTx are not known. While LD is cheaper than DD in kidney transplantation, the 2 year cost analysis after kidney transplant is comparable between LD and DD [34]. Since UTx is a transient transplant and immunosuppression is stopped following eventual removal of the graft, lifetime costs are lower in uterus transplant as compared to other solid organ transplants. The average cost for UTx in a European setting is about €74,564 (€42,984 for recipient and €31,580 for donor) [35]. This includes sick leave, preoperative investigation, IVF, anesthesia, medications, post-operative care, and re-hospitalizations for donor and recipient. There has not been published research on the cost in the United States.

#### 2.3.4. Transgender Considerations

As the field of UTx matures, this quality-of-life enhancing transplant will eventually be offered to more people without functional uteri, such as individuals with androgen insensitivity syndrome and transgender individuals. To date, there has not been a uterus transplanted in a transgender female, and the possibility remains theoretical. Many have argued that there is not an ethical reason to reject a trans patient from receiving a uterus transplant, as a trans patient does not inherently have less claim to desiring pregnancy than their cis-female counterparts [36]. There have been recent calls to action to include the trans community as a possible recipient population for UTx. Use of UTx in cis-males has similar challenges, and the ethical and social considerations are still being debated [37].

It is not known whether the LD or DD approach is better for these patients. However, it has been suggested that the hysterectomy required for uterus procurement for a trans patient would necessitate use of DDs, as a large portion of the donor vagina and accompanying vasculature can be removed from a DD [17]. More research is needed with regards to the transplant surgery, hormone treatment, and pregnancy in this population.

## 3. Conclusions

Equipoise is a principle in research whereby certain medical interventions are justified if there is sufficient balance between potential benefits and potential harms [38,39]. In transplant surgery, where both the donor and recipient must be accounted for when weighing the risks of the intervention, the term “double equipoise” has been introduced [40]. To fully address equipoise in the content of LD vs. DD models of UTx, we must be able to compare long-term outcomes for donors and recipients. However, there is still a paucity of data of long-term health outcomes for recipient and subsequent offspring. In the short term, while the DD model obviates donor risk, it is not clear that outcomes for the recipient are similar or superior, despite theoretical benefits of being able to harvest alternate or more generous vascular pedicles. Furthermore, the benefits of the DD model must be weighed against the shortage of available DDs.

The first successful uterus transplantation was performed as part of a carefully planned clinical trial in Sweden. While that livebirth marked the beginning of a new era in reproductive surgery, the event was also an ending point, a culmination of many years of preparation and multiple demonstrations of successful transplantation in animal models. For subsequent uterus transplant programs that did not perform a similar stepwise approach with animal models, these programs relied on close mentorship with established programs.

A similar stepwise approach for a uterus transplant center can apply to the use of LDs and DDs, where initial exclusive use of DDs gradually expands to include LDs as ability and expertise increases. As uterus transplantation becomes more routine and the safety and efficacy of LD procurement is further established, the concern for risk to LDs lessens. Such an approach could theoretically reduce the relatively high complication seen in early trials of LD UTx.

Ultimately, centers that utilize both LDs and DDs will allow researchers to better compare long-term outcomes between LD and DD models within a similar recipient population utilizing the same team of surgeons and other specialists.

In conclusion, while an LD model is appealing—it is easier to implement, does not rely on the unpredictability of DD organ availability, provides more authentic consent and detailed medical history, among the other advantages discussed here—the complexity of the procurement procedure presents serious and real risks to the living donor, with key differences in directed and non-directed donors. For the individual desiring to donate their uterus, a thorough understanding of the known risks is mandatory; while their own autonomy to perform supererogatory acts should be respected, this ethical appeal can only be acceptable in the setting of sufficient informed consent. For the center desiring to begin a uterus transplant program, a thorough understanding of the procedure, processes, and ethical considerations is likewise critical.

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
