# Peer review of "Uterus Transplantation: Revisiting the Question of Deceased Donors versus Living Donors for Organ Procurement"

_jcm, 2022, doi:10.3390/jcm11154516_

Round 1

Reviewer 1 Report

I would like to thank authors for this paper. It summarizes the most important information about living- and deceased uterine donors and their utility for uterus transplantation. This paper is very well-written. It does contain most of the crucial information. I do have only one major objection, i.e., I feel that the paper does not cover sufficiently the most important aspect, which is the comparison of the outcome of UTx with LD vs. DD grafts. In fact, the short paragraph is way too short and gives few information. In fact, this should be the core of the paper, not so much the section on the ethical considerations. I am aware that there is a paper to be published in JAMA Surgery, but I think that more information should be given on the comparison of the outcome of both sources of grafts (live-birth rate per attempted transplant, live-birth rate per technically successful transplant, mean number of embryo transfers needed for pregnancy, mean time from UTx to pregnancy, number of embryo transfers/pregnancies/miscarriages/live-births, etc.). All other information is well-known. It is important to know what type of grafts yields better outcomes with respect to primary aim achievement (live birth).

Minor objection:

See lines 54-69 - why is there the identifier not given by all of the trials mentioned?

See lines 174-175 - why would LD be not ethically permissible for a LD in premenopausal age? Please could you provide a brief explanation? I don't see any reason why mother of a child/children who is premenopausal and does not wish to have any more children should not donate her uterus. Or donor nulliparous woman who does not wish to bear children. In my opinion, accepting menopausal grafts often could mean accepting marginal grafts.

Overall, I congratulate the authors to the well-written comprehensive paper on advantages of both types of uterine grafts.

Author Response

Point 1: I would like to thank authors for this paper. It summarizes the most important information about living- and deceased uterine donors and their utility for uterus transplantation. This paper is very well-written. It does contain most of the crucial information. I do have only one major objection, i.e., I feel that the paper does not cover sufficiently the most important aspect, which is the comparison of the outcome of UTx with LD vs. DD grafts. In fact, the short paragraph is way too short and gives few information. In fact, this should be the core of the paper, not so much the section on the ethical considerations. I am aware that there is a paper to be published in JAMA Surgery, but I think that more information should be given on the comparison of the outcome of both sources of grafts (live-birth rate per attempted transplant, live-birth rate per technically successful transplant, mean number of embryo transfers needed for pregnancy, mean time from UTx to pregnancy, number of embryo transfers/pregnancies/miscarriages/live-births, etc.). All other information is well-known. It is important to know what type of grafts yields better outcomes with respect to primary aim achievement (live birth).

Response 1: We thank the reviewer for their kind words and helpful feedback. The Outcomes data section has been greatly expanded to include many of these points, when possible (Section 2.3.1 Lines 241-287).

Point 2: Minor objection:
See lines 54-69 - why is there the identifier not given by all of the trials mentioned?

Response 2: We thank the reviewer for bringing this to our attention. All clinical trial identifiers are now included (Lines 66-72).

Point 3: See lines 174-175 - why would LD be not ethically permissible for a LD in premenopausal age? Please could you provide a brief explanation? I don't see any reason why mother of a child/children who is premenopausal and does not wish to have any more children should not donate her uterus. Or donor nulliparous woman who does not wish to bear children. In my opinion, accepting menopausal grafts often could mean accepting marginal grafts.

Response 3: We thank the reviewer for their attention to this point. This was an oversight. We have adjusted the sentence from “would not be ethical” to it would be “would be more ethically problematic”. Removal of the uterus in a premenopausal women via a radical surgery puts her ovarian function at risk (hastening time to menopause and increasing risk of ovarian failure) as well as increases risk of regret (Line 222).

Point 4: Overall, I congratulate the authors to the well-written comprehensive paper on advantages of both types of uterine grafts.

Point 4: We appreciate the reviewer’s attention and helpful feedback. Thank you.

Reviewer 2 Report

The authors present the reasons for LD vs DD uteri for purpose of transplantation. The concerns are solely of organ source and not other issues pertaining to uterine transplantation for AUI.   Comments

a.   section 2.1.3:  It is not clear why this is included in favor of LD approach.  Certainly the LD must provide consent to undergo the procedure.  The elements of what information must be provided to the potential donor should be distinguished from what is told the "authorizing" individual of the DD uterus.  The concept of "informed consent" should probably not be applied to DD organs or tissues.  "Authorization" to donate either organ, VCA or tissue is a more accurate term.  Virtually no OPO will obtain obtained true informed consent (from family) prior to retrieval of any of the medical products of human origin.  Informed consent from a LD vs authorization of the uterus donation (DD) should be discussed.  Whether first person designation for organ donation is sufficient to uterus/VCA donation may be debatable, but the authors present some reasons why family approach may be beneficial.  May want to rewrite this section a bit.

b. The authors may want to address "learning curve" for complications to be discussed with donor and center experience.  This was widely demonstrated in liver live donation.

c. are there technical differences in LD vs DD uterus txp: vascular pedicles or vaginal cuff that should be discussed.  Certainly in liver and kidney transplantation, there are fewer technical options options in LD grafts and reconstructions tend to be more more complex with LD grafts.   These should be discussed.

d. section 2.2.4 line 170 ff: it should be clarified that the risk  of taking estrogens is to the donor (recipient of the hormone) and not the recipient of the uterus.

Author Response

The authors present the reasons for LD vs DD uteri for purpose of transplantation. The concerns are solely of organ source and not other issues pertaining to uterine transplantation for AUI.   Comments

Point 1:  section 2.1.3:  It is not clear why this is included in favor of LD approach.  Certainly the LD must provide consent to undergo the procedure.  The elements of what information must be provided to the potential donor should be distinguished from what is told the "authorizing" individual of the DD uterus.  The concept of "informed consent" should probably not be applied to DD organs or tissues.  "Authorization" to donate either organ, VCA or tissue is a more accurate term.  Virtually no OPO will obtain obtained true informed consent (from family) prior to retrieval of any of the medical products of human origin.  Informed consent from a LD vs authorization of the uterus donation (DD) should be discussed.  Whether first person designation for organ donation is sufficient to uterus/VCA donation may be debatable, but the authors present some reasons why family approach may be beneficial.  May want to rewrite this section a bit.

Response 1: We thank the reviewer for their feedback and agree that this wording needed to be altered. We have made the change from family consent to family authorization for DD procurement (Lines 113, 118) and exapanded our discussion on the consent process (Lines 128-130).

Point 2: The authors may want to address "learning curve" for complications to be discussed with donor and center experience.  This was widely demonstrated in liver live donation.

Response 2: We thank the reviewer for their suggestion. We address the concept of a learning curve in our conclusion, but we agree that the manuscript would benefit from additional discussion of this point. We have added discussion of the “learning curve” in living donor liver transplantation in the section “Avoidance of risk of physical harm to the donor” (Lines 171-175).

Point 3:  Are there technical differences in LD vs DD uterus txp: vascular pedicles or vaginal cuff that should be discussed.  Certainly in liver and kidney transplantation, there are fewer technical options in LD grafts and reconstructions tend to be more complex with LD grafts.   These should be discussed.

Response 3: We thank the reviewer for their feedback. This important point is discussed in section 2.2.2. We have added the reviewer’s point about liver and kidney transplantation having greater technical complexity in LDs compared with DDs (Lines 189).

Point 4:  section 2.2.4 line 170 ff: it should be clarified that the risk  of taking estrogens is to the donor (recipient of the hormone) and not the recipient of the uterus.

Response 4: We thank the reviewer for pointing out this error. This has now been corrected (Lin